# Trimetallic Nanocomposites Grafted on Modified PET Substrate Revealing Antibacterial Effect Against *Escherichia coli*

**DOI:** 10.3390/molecules30244820

**Published:** 2025-12-18

**Authors:** Veronika Lacmanová, Veronika Svačinová, Martin Petr, Petr Slepička, Filip Průša, Ondřej Kvítek, Anna Kutová, Alena Řezníčková, Karolína Šišková

**Affiliations:** 1Department of Solid State Engineering, University of Chemistry and Technology Prague, 166 28 Prague, Czech Republicpetr.slepicka@vscht.cz (P.S.); ondrej.kvitek@vscht.cz (O.K.); kutovaan@fzu.cz (A.K.); 2Department of Experimental Physics, Faculty of Science, Palacký University Olomouc, 17. Listopadu 12, 779 00 Olomouc, Czech Republic; 3Regional Center of Advanced Technologies and Materials, CATRIN, Palacký University Olomouc, Šlechtitelů 27, 779 00 Olomouc, Czech Republic; martin.petr@upol.cz; 4Department of Metals and Corrosion Engineering, University of Chemistry and Technology Prague, 166 28 Prague, Czech Republic; filip.prusa@vscht.cz

**Keywords:** antibacterial surface, nanostructures, *Escherichia coli*, polymer substrate, plasma, grafting

## Abstract

The microbial resistance era prompts researchers to find new effective antimicrobial agents. Trimetallic nanocomposites consisting of AuAg nanostructures and iron oxide nanoparticles can represent an efficient tool to inhibit bacterial growth as demonstrated here. The trimetallic nanocomposites, prepared by green chemistry approach, are grafted on a modified, plasma-treated, flexible polyethylene terephthalate (PET) substrate. Spectroscopic and microscopic characterization of the trimetallic nanocomposites grafted on modified PET together with antibacterial tests confirm the successful applicability of the newly developed material: a statistically significant antibacterial effect against *E. coli* is proven. This effect is further pronounced by a short time (5 min) UVA light (365 nm) irradiation. The present work thus reports on the feasible preparation of the brand-new material that is successfully used in *E. coli* colony growth regulations. The impact of small noble metal nanostructures containing Ag and UVA light-activated iron oxide particles on the bacterium can be combined and results in the improved antibacterial performance of the final material. Employing such material may represent a potential strategy for fighting against the development of bacteria resistance.

## 1. Introduction

Polymer materials represent indispensable components of our daily lives; plastic bottles made of polyethylene terephthalate (PET) are one of the examples. Polymers are susceptible to bacterial attachment and consequent biofilm formation [1,2,3,4], especially when used for a longer period. The situation starts to be severe owing to an augmenting bacterial resistance to antibiotics. Hence, there is an urgent need to develop new surface coatings, enabling either a complete suppression of biofilm formation (bactericidal effect), or an effective regulation of bacterial growth (antibacterial effect).

To tackle the known multidrug resistance of bacteria, researchers have developed different kinds of metal-based nanomaterials and nanocomposites [5,6,7,8,9,10,11,12,13,14,15,16]; the terms metalloantimicrobials [9] and/or metalloantibiotics [10] have been coined recently. These nanomaterials and nanocomposites revealed antimicrobial properties, mostly due to higher surface-to-volume ratio and their small size, enabling easier penetration through cell membrane. Therefore, grafting of polymer surface with metallic and/or oxidic nanoparticles/nanostructures containing metal elements, such as Ag, Cu, Zn, Ti, etc., [17,18,19] manifested itself as a prospective and intriguing approach.

Raymond Turner in his recent review [9] exhorts to the responsible usage of metalloantimicrobials and suggests the usage of antimicrobial mixtures that target the cell in different ways to avoid easily evolved bacteria resistance. In fact, it has already been shown that the potency of selected metal elements within nanostructures serving as antibacterial agents can be improved by alloying, e.g., Ag with Pt at gold nanorods [20]. Therefore, the combination of several metals (two or even three) within one nanocomposite represents a new trend of research nowadays.

Moreover, bimetallic and trimetallic nanocomposites can reveal synergistic antimicrobial activity. As an example, trimetallic Au/Pt/Ag nanoparticle-based fluids exhibiting better antibacterial properties than bimetallic and single-metallic nanofluids [21] can be mentioned. In addition to nanofluids, trimetallic Au/Pt/Ag nanoparticles synthesized by bio-extracts have been recognized as excellent antimicrobial agents against planktonic and sessile clinical strains [22]. In another work [23], trimetallic nanocomposites containing Ag, Cu, and Pb are shown to possess antibacterial activity against *Escherichia coli*. Simultaneously, these trimetallic nanocomposites demonstrated cell toxicity towards potential disease-causing cells (human nasopharyngeal epithelial cancer cell line) [23].

Other types of nanocomposites compose of three metallic oxides and are employed as antimicrobial agents, which has been reported in the scientific literature as well. For instance, CuO-NiO-ZnO trimetallic oxide nanoparticles showed a detrimental effect on two major bacterial strains [24]. Furthermore, biosynthesized trimetallic oxide Cu/Cr/Ni nanoparticles exhibited the inhibition of bacterial growth [25].

Based on the above-mentioned points, the combination of suitable metal/s and metal oxide particles can potentially lead to the development of a new type of antibacterial coating. For instance, the unique combination of Au nanostructures and photo-activable iron oxide particles, both embedded in a protein matrix and then chemically attached to a PET substrate, has been tested against *E. coli* very recently [26]. A slight antibacterial effect was observed in the final material, consisting of bimetallic nanocomposites grafted on modified PET substrate [26].

Here, we would like to employ AuAg nanostructures combined with iron oxide particles within one nanocomposite (to be referred to as the trimetallic nanocomposite and abbreviated as AuAg-BSA-SPION), chemically attach these trimetallic nanocomposite on modified PET substrates, and then test the antibacterial properties of the final material (AuAg-BSA-SPION@pl-BPD-PET). The first novelty of the present work lies in the preparation of the final material per se. In the next step, we want to evaluate the effect of the final material against *Escherichia coli* for two reasons: (i) it has been proven that hydroxyl radicals are effective in *E. coli* growth inhibition [27], and (ii) the results can be compared with the previous work dealing with grafted bimetallic nanocomposites serving as antibacterial PET coatings against *E. coli* [26]. It is expected that due to the presence of Ag within the trimetallic nanocomposites, the final material will provide a more pronounced antibacterial effect than that obtained for the case of the grafted bimetallic nanocomposites (Au-BSA-SPION, our recent work [26]) because it is known that inhibitory action against Gram-negative bacteria is higher for nanosilver than for any other metallic nanoparticles [28].

According to the literature [26,29], the optimized plasma treatment of PET (representing a flexible, light, UVA-transparent polymeric substrate) has been selected to change its inert character. The optimal procedure of plasma treatment can not only activate the surface of PET, while its negligible degradation occurs, but can simultaneously remove any dirtiness from the PET surface. Subsequently, a dithiol linker (BPD) could be employed to graft the trimetallic nanocomposites on the modified PET surface. From one of our previously published works [30], we know that BPD represents a “rigid” linker and as per se, it is conveniently oriented on the PET surface with one of its two sulfhydryl groups pointing outside the surface. This can enable noble metal nanostructures to bond via the covalent interaction between Au/Ag with S, as evidenced for AuNPs and/or AgNPs in refs [31,32].

Importantly, the trimetallic nanocomposites grafted in this work were synthesized by the green chemistry (protein-templated) approach that has been developed by us recently [33]. The trimetallic nanocomposites consist of two kinds of functional nanostructures: AuAg irregularly shaped nanoclusters (diameters below 2 nm) embedded within the protein (BSA) and SPION (superparamagnetic iron oxide nanoparticles of sizes well below 8 nm) attached to the same protein [33]. These two kinds of nanostructures are positioned at different binding sites of the protein since Au and Ag prefer the interaction with sulfhydryl groups (mostly cysteine residues), while iron binds to N-terminal region of the protein [34] and may prefer to interact with carboxylates, oxygen, and/or nitrogen-terminated functional groups. Therefore, we expect that AuAg nanostructures embedded in BSA matrix can be bonded via the “free” sulfhydryl group of BPD to PET substrate, enabling then the attachment of the whole trimetallic nanocomposite. Consequently, the steric arrangement of the trimetallic nanocomposite can change while grafting to the surface, leading to new properties that have not been observed so far, e.g., antibacterial properties.

Moreover, it is known from the literature [35] that the binding mode is one of key determinants for the antimicrobial performance of iron oxide/silver nanocomposites. It should be noted that the trimetallic nanocomposites, developed by us [33], revealed biocompatibility with healthy cell lines (RPE-1) despite the presence of Ag (till 0.12 mg/mL). Nevertheless, the biocompatibility of these trimetallic nanocomposites towards cells may change due to potential conformational changes in protein matrix, induced because of their grafting on PET substrate. Furthermore, it is important to investigate the impact of UVA light exposure for short times (5 min) on the antibacterial properties of the final material because of the presence of SPION (as inspired by the recent and previous works [26,36]). However, any prolonged UVA-light irradiation should be avoided because it is known to cause irreversible changes to the nanocomposites containing noble metal nanostructures embedded in BSA [37]. Therefore, a direct comparison of antibacterial properties of the final material under no light vs. UVA light irradiation represents another novelty of this work.

## 2. Results and Discussion

We developed a new material consisting of trimetallic nanocomposites (AuAg-BSA-SPION) grafted on plasma-treated, BPD-modified PET substrates (pl-BPD-PET). The schematic depiction of the final material (AuAg-BSA-SPION@pl-BPD-PET) formation is shown in Figure 1.

Since detailed characteristics of trimetallic nanocomposites per se are published in our previous work [33], in this paper, we focused our attention on the characterization of the final material, AuAg-BSA-SPION@pl-BPD-PET. In the following sections, we are going to present the results of the spectroscopic and microscopic characterization of the newly developed material (AuAg-BSA-SPION@pl-BPD-PET). In the subsequent section, the wettability of the final material, as well as that of the flexible PET substrate, after each step of its surface modification (i.e., after plasma treatment and after BPD attachment) is evaluated by goniometric water contact angle measurements and mutually compared. It is tremendously important to know the aqueous wettability of the final material surface because it represents an essential factor influencing antibacterial results. Last but not least, antibacterial effect against *Escherichia coli* without and under UVA-light irradiation (365 nm for 5 min) is tested and the results are discussed.

### 2.1. Spectroscopic Characterization of AuAg-BSA-SPION@pl-BPD-PET

Several spectroscopic techniques were used to verify if the characteristics of modified PET substrate were changed after the grafting of the trimetallic nanocomposites. First, absorption spectroscopy in the visible region was employed as a common laboratory spectroscopic method of sample characterization. The visible spectrum of PET substrate modified by plasma treatment and BPD (pl-BPD-PET) is compared with that of the modified PET substrate grafted with trimetallic nanocomposites directly (Appendix A). The two bands positioned within the visible spectrum of pl-BPD-PET (with the maxima at around 460 and 510 nm—Appendix A) are slightly broadened in the spectrum of trimetallic nanocomposite grafted on this substrate. Moreover, the whole spectrum of the final material is “sitting” on a background that is most probably stemming from the presence of the trimetallic nanocomposites.

Middle range IR absorption spectroscopy, representing a useful tool of vibrational spectroscopy, was employed to investigate functional groups of prepared materials. It should be noted that IR absorption spectrum is a bulk material characteristic; however, potential changes within the spectral pattern could be detected even for surface-modified samples. Hence, IR absorption spectra of PET substrate, modified PET substrate (Ar plasma-treated + BPD-modified), and the final material (AuAg-BSA-SPION@pl-BPD-PET) were recorded, as shown in Appendix A. With respect to the fact that only surface modifications of PET substrate (which provides a very intensive IR signal by itself) were performed, only minor changes within the IR absorption patterns were observed, namely slight shifts in band maxima and changes in relative bands intensity. From the direct comparison of IR spectra of PET and that of PET-pl-BPD (Appendix A), it is notable that the band positioned at 1240 cm^−1^ is shifted to 1236 cm^−1^, and 1095 to 1092 cm^−1^, and the relative intensity of the 1236, 1092, and 1712 cm^−1^ bands decreases. The latter three bands can be attributed to the vibrations of PET ester group. This suggests that the modifying species (BPD) are interacting with the ester groups of PET. The same changes in IR absorption pattern are further pronounced in the IR spectrum of the final material (AuAg-BSA-SPION@pl-BPD-PET), as seen in Appendix A: the observed band maxima shifted further from 1092 to 1086 cm^−1^, 1236 to 1233 cm^−1^, and 1712 to 1708 cm^−1^. Concomitantly, a further relative intensity decrease was detected. The shifts in band maximum positions and decrease in the relative intensity of the three above-listed bands report about an attenuation of PET functional group vibrations due to the presence of bulky modifying species/nanocomposites at the surface of PET substrate.

Although visible and IR absorption spectroscopies gave indirect evidence about the presence of trimetallic nanocomposites at modified PET substrates, we investigated the surface by means of XPS which provides more accurate results and surface composition (basically from the most upper 10 nm layer of the sample) than the two above-mentioned absorption spectroscopies. XPS of the final material were recorded at two selected angles: 0° (i.e., perpendicular to the surface, providing a general sample composition below the surface) and 81° (the angle measured from the normal to the surface; more information is provided directly from the top surface layers), as shown in Figure 1A and Figure 1B, respectively. For the sake of a direct comparison, the XPS signal of the trimetallic nanocomposite per se, deposited by drop-casting method on a Si wafer (regularly used in XPS spectroscopy for deposition of liquid samples) and measured at 0° (perpendicular to the surface), is shown in Figure 1C.

Based on survey scans and XPS signal analyses, the atomic percentage of selected elements can be determined and is listed in Table 1 for the trimetallic nanocomposite dropped and/or grafted on substrates.

Evidently, the presence of C, O, N, S, Fe, Ag, Au was confirmed by XPS measurements in the final material (Table 1, Figure 1A,B). All these elements are present in trimetallic nanocomposite itself. Nitrogen can even be regarded as a marker of BSA because it is not included in the chemical formulae of PET, BPD (where solely C, O, and S are involved), and Si substrate. Hence, the nitrogen content confirms the trimetallic nanocomposite presence directly.

Surprisingly, iron was not detected by XPS in trimetallic nanocomposite per se when the drop-deposition technique was used—Table 1 and Figure 1C—although Fe presence was well-documented by ICP-MS (inductively coupled plasma mass spectrometry). The concentrations of each metal derived from ICP-MS measurements are as follows: 0.32 mg/mL of Fe, 1.01 mg/mL of Au, and 0.16 mg/mL of Ag [33]. To validate the missing XPS signal of Fe in the trimetallic nanocomposite when drop-casted, the same deposition technique was applied on two types of substrates: Si wafer (hydrophilic as demonstrated in [38]) (Figure 1C) and pl-BPD-PET (hydrophobic [17]) (Appendix A). In both cases, regardless of their hydrophilicity/hydrophobicity, the small drop of trimetallic nanocomposite aqueous solution deposited several times is allowed to quickly evaporate (within half-an-hour) to enable XPS analysis.

Based on the results, it can be stated that there are several interrelated factors influencing the final XPS signals: (i) rearrangement of the trimetallic nanocomposite inner structure occurs near the surface; (ii) bonding types of the nanocomposite towards Si wafer and pl-BPD-PET substrates substantially differ (noncovalent vs. covalent, respectively); (iii) differences in thicknesses and distribution of trimetallic nanocomposites on substrates should also be considered. Indeed, there is a rather thick and uneven layer of trimetallic nanocomposites on substrates when drop-casting and fast solvent evaporation are applied. On the contrary, thin layer and relatively even distribution on pl-BPD-PET substrates can be observed due to covalent bonding of the trimetallic nanocomposite via BPD and the longer time (24 h) provided to its rearrangement near the surface.

From the above discussion, one can hypothesize that the drop-casting of trimetallic nanocomposite and subsequent solvent evaporation may lead to differences in preferential orientation of hydrophilic and hydrophobic parts of the protein matrix during the process of solvent evaporation. This idea is further supported by a huge discrepancy not only in Fe 2p XPS signal, but also in C 1s, O 1s, N 1s, S 2p, and Au 4f XPS signals, as well as the derived percentage contents when comparing trimetallic nanocomposite drop-casted vs. grafted (both measured at 0°)—see Table 1 (first three columns). While C, N, and Au dominates in drop-casted trimetallic nanocomposites, their contents significantly decrease in the final material with simultaneous increase in the XPS signal stemming from O, Fe, and Ag (note: the increased content of S is not discussed in this moment because sulfur is included within BPD structure; hence, it contributes to the overall S percentage)—Table 1. Assuming the character of the XPS signal (coming from 1 to 10 nm below the measured sample surface) and the hydrodynamic diameter of our trimetallic nanocomposite (hundreds of nanometers as determined by dynamic light scattering in our previous work [33]) that may be further increased by nanocomposite aggregation on substrates (induced by solvent evaporation), we hypothesize that the XPS signal from the trimetallic nanocomposite drop-casted on a Si wafer stems mostly from the hydrophobic parts that do not contain a detectable amount of iron. Therefore, iron is not detected by XPS in trimetallic nanocomposite per se (i.e., its content is below the detection limit of XPS). Contrary to the drop-casting method of XPS sample preparation employed in the trimetallic nanocomposite case, covalent bonding via HS- groups of BPD is assumed to take place in the case of the final material (AuAg-BSA-SPION@pl-BPD-PET). Rearrangement of the protein matrix appears. Consequently, XPS signal of Fe is detected in the final material and atomic percentage exceeds 2%—Table 1.

Let us now compare the XPS signals of the final material recorded at two different angles which enables us to determine the composition of layers below the surface (0°) vs. top layers (81°)—Table 1 (the last two columns on the right). The atomic percentage of Fe increased, together with a slight increase in C and O content in the top surface layers, while, on the contrary, that of S, N, Ag decreased, and Au was preserved on the same value (Table 1). Based on these results, one can deduce that AuAg nanoclusters are preferentially interacting with BPD (as it has been assumed—Figure 1), while, on the other hand, SPION are more exposed to the ambient environment, i.e., located dominantly at the top layers of the final material.

XPS results thus report about the uneven vertical distribution of different types of nanostructures (AuAg vs. SPION) embedded within the protein matrix at the surface of the modified PET substrate. It should be pointed out, however, that both nanostructures (AuAg and SPION) are embedded in the same protein matrix (BSA), thus creating a compact nanocomposite. Moreover, preferences in the orientation of the protein scaffold at the modified PET substrate should be considered as well. Therefore, it is not always feasible to orient a particular nanoparticle toward its preferred environment. This explains why the changes in the content of Au, Ag, and Fe are not solely related to their bonding preferences and lead to the observed uneven vertical distribution. Nevertheless, their preferential location within the final material can be still derived from XPS data. Importantly, several AuAg nanostructures (roughly two thirds, as derived from the XPS quantification of survey scans—Table 1) can be exposed to the ambient environment (Figure 1).

More detailed XPS investigation of selected regions of interest, such as Fe 2p, Au 4f Ag 3d, and S 2p (Figure 2), recorded at two angles 0° (information from subsurface layers) and 81° (information from top layers), provided evidence that metal atoms are preferentially present in their oxidized forms. Iron is present solely as Fe (III) regardless of the surface layer (Figure 2A,B). It is in accordance with our previous investigation [33] of these trimetallic nanocomposites made by Mossbauer spectroscopy (which is a bulk iron-sensitive technique). On the contrary, the percentage of oxidized forms of Au and Ag differ in subsurface vs. top layers: approx. 67.5% of Au (δ+) and 65% of Ag (I) are detected in subsurface layers (Figure 2C,D), which is further increased to 79% of Au (δ+) and 72% of Ag (I) in the top layers (Figure 2E,F). Intentionally, we assigned oxidized Au as Au (δ+) because not only Au (I), but also Au (II), can be present as we know from our previous studies using electron paramagnetic resonance (EPR) spectroscopy [37]. It can thus be concluded that the Au and Ag atoms contained within top surface layers of the final material (exposed to the ambient environment) are prone to oxidation: Au by 11.5%, while Ag by 7%. Inspecting the S 2p region in detailed XPS scans recorded at two different angles gave evidence about an increased Au-S percentage content (by approx. 13%) in the top surface layers at the expense of R-SH and sulfate—compare Figure 2G and Figure 2H. This roughly correlates with the increased Au (δ+) percentage content discussed above.

It may be surprising that gold, considered as a more noble metal, is oxidized to a higher extent than silver. However, it can be envisaged that there are changes within the nanocomposite inner structure during the process of grafting. Therefore, comparing percentages of Au and Ag oxidation states in the trimetallic nanocomposites prior and after grafting (both measured at 0°—compare Appendix A and Appendix A vs. Figure 2C and Figure 2D, respectively), there are significant differences. Indeed, the content of oxidized forms of Au and Ag in the final material (AuAg-BSA-SPION@pl-BPD-PET) contradicts their percentage in the trimetallic nanocomposite itself when drop-deposited on a Si wafer where 72.5% of Au (0) and 27.5% of Au (δ+)—Appendix A, and 100% of Ag (I); Appendix A— were detected (note that the very similar contents of Au (0), Au (δ+), and Ag (I) were determined by XPS for AuAg-BSA nanoclusters in our very recent work [39]). Obviously, a big portion of gold atoms (roughly 40%) is oxidized when the trimetallic nanocomposite is grafted to pl-BPD-PET, i.e., Au (δ+) content increased, while, simultaneously, Au (0) content decreased by approx. 40%—Figure 2C. This may be explained by covalent bond formation between the outer shell of noble metal nanostructures and HS-groups stemming from BPD. Surprisingly, 35% of Ag atoms within the trimetallic nanocomposite is reduced to zero valence oxidation state upon AuAg-BSA-SPION grafting to pl-BPD-PET substrate—Figure 2D. Hence, it reveals that an anti-galvanic process is involved during trimetallic nanocomposite grafting to pl-BPD-PET. It might be hypothesized that the inner structure of AuAgNCs was changed during grafting, i.e., the positions of Ag and Au reorganized within the AuAgNCs in a way that around one third of Ag atoms migrated toward the interior of the noble metal nanostructures, becoming Ag (0), while Au atoms are exposed to outside, enabling grafting to pl-BPD-PET via Au-S bond formation (thus, increased Au (δ+) in the Au 4f region and a concomitant increase in the Au-S in S 2p region of XPS detailed scans were observed). Nevertheless, further experiments using more advanced experimental techniques, such as those exploiting synchrotron radiation, would be necessary to confirm this hypothesis.

### 2.2. Microscopic Characterization of AuAg-BSA-SPION@pl-BPD-PET

Scanning electron microscopic (SEM) imaging was performed for PET substrate, PET-pl-BPD sample, and the final material; their representative images are shown in Figure 3.

Since the resolution of SEM is insufficient for the used nanostructures, we decided to measure atomic force microscopy (AFM). In Figure 4A–C, there are not only AFM images of the representative samples of PET substrate, the modified PET substrate (pl-BPD-PET), and the final material (AuAg-BSA-SPION@pl-BPD-PET), but also values of surface roughness (Ra) determined for 3 × 3 µm^2^ areas of the AFM images. Moreover, 3D AFM images and surface roughness derived from 300 × 300 nm^2^ areas are shown and compared for the modified PET substrate and the final material in Figure 4D,E.

In accordance with the literature (e.g., [30,31]), surface roughness increases, going from PET substrate (Ra = 0.5 nm, meaning smooth surface) to plasma-treated BPD-modified PET (Ra = 1.7 nm). Grafting trimetallic nanocomposites on pl-BPD-PET substrate leads towards a further surface roughness increase (more than twice) as determined for the final material (Ra = 3.6 nm). Surface roughness measurements thus proved a successful grafting of trimetallic nanocomposites on the modified PET surface.

Considering AFM images (Figure 4), surface structures of the final material seem to be globular, rounded, without presence of any sharp crystals and/or edges. Consequently, it could be suitable for cell adhesion. In fact, surface roughness represents one of the factors influencing the adhesion of cells, in general, and bacteria, in particular. Very smooth surfaces are inconvenient for cell adhesion, while too rough surfaces may cause deformation of cell walls due to adhesion into valleys [40]. On the other hand, cell adhesion is also influenced by surface aqueous wettability [40]; therefore, it was investigated by us as well, and the results are discussed in the next section.

### 2.3. Goniometric Contact Angle Measurements

Static water contact angle (WCA) of distilled water was dropped on the surface of (i) PET, (ii) substrates in each step of their modification (Ar plasma; Ar plasma + BPD), and (iii) the final material was evaluated. The results are visualized and compared in Figure 5.

WCA of PET substrate reached the value of around 83° (Figure 5A). The wettability of PET substrates decreased by the subsequent Ar plasma treatment, as well as by BPD modification (WCA values above 96°)—Figure 5B,C. On the contrary, WCA decreased to the value of 66.6° ± 0.6°, while trimetallic nanocomposites grafted on modified PET substrate (Figure 5D). The WCA values obtained for the final material (AuAg-BSA-SPION@pl-BPD-PET) are comparable with those published for other systems, where Ag nanoparticles of different origins are grafted on activated PET substrates [17]. Therefore, it can be concluded that the wettability of the final material developed within this work improved in comparison to that of pristine and/or modified PET substrates. Due to the presence of a tiny amount of Ag, its potential usage in bacteria growth regulation could be envisaged.

### 2.4. Antibacterial Effect Against Escherichia coli

Potential antibacterial properties of the final material developed within this work (AuAg-BSA-SPION@pl-BPD-PET) were investigated, employing the colony-producing/forming assay (i.e., drop-plate method). For the sake of a direct comparison, the same tests were performed for PET substrate, plasma-treated PET, and plasma-treated BPD-modified PET (pl-BPD-PET) substrate simultaneously. As a control (CTRL), a diluted bacterial suspension deposited at the bottom of a sterile Petri dish was used. The measurement was carried out using the Gram-negative bacteria *Escherichia coli* to compare the results with the previous work [26]. The results are presented as column graphs in Figure 6.

Considering the decrease in CPU (colony-producing units) average number in the case of the final material when compared to the control (a decrease from 32.5 CPU to approx. 25 CPU—compare black vs. orange columns in Figure 6, left part), it can be stated that a mild, but statistically significant, antibacterial effect of the final material against *E. coli* is observed. This mild antibacterial effect can be beneficial for the regulation of *E. coli* growth. PET substrate and/or modified PET substrates did not provide such antibacterial responses (Figure 6, left part). In fact, their antibacterial effect is negligible and falls within the experimental error (red, blue, and green columns in Figure 6, left part).

Since we do not expect antibacterial activity against *E. coli* to be induced by the protein (BSA), neither by eco-friendly-prepared Au nanostructures [41] and/nor SPION present within the final material [26], the antibacterial effect is most probably induced by the presence of Ag (despite its very low concentration at the surface, 0.2% derived from XPS data, Table 1). Ag nanoparticles and released Ag ions are known to interact with biomolecules, consequently resulting in a detrimental effect on bacteria [42]. Additionally, radical oxygen species formation induced by Ag nanoparticles may be considered as another mechanism of its antibacterial action [42,43]. The assumption about the negligible impact of Au, SPION, and BSA on *E. coli* growth can be well-documented by referring to our previous work dealing with Au-BSA-SPION composites grafted on surface-modified PET substrate (Au-BSA-SPION@pl-BPD-PET) [26]. Albeit there are scientific articles about bactericidal effect of Au nanoparticles on *E. coli* [44], the decrease in Au size to nanoclusters (below 2 nm in diameter) and their entrapment in the protein matrix may be reasons for there being no antibacterial effect of Au-BSA-SPION@pl-BPD-PET on *E. coli* [26,41].

It is quite hard to directly compare the results of antibacterial testing of the final material under study with any other three-metal composite published so far in the scientific literature because different methods of antibacterial effect evaluation were employed, such as agar disk diffusion test [21], well diffusion assay [23], and macro broth dilution approach^25^, for instance. The only available article using the drop-plate method and reporting on antibacterial properties against *E. coli* has dealt with CuO-NiO-ZnO nanoparticles prepared by a co-precipitation method [24]. Considering the (i) tested concentration range (1–10 mg/mL) of CuO-NiO-ZnO nanoparticles and the (ii) investigated time of bacterial growth (2–11 h), these three-metal oxidic nanoparticles [24] revealed a negligible antibacterial effect in comparison to the final material under study in the present work. Indeed, their [24] lowest concentrations of CuO-NiO-ZnO (1 mg/mL), when allowed to interact with *E. coli* for 11 h, manifested itself by an approx. 14% decrease in E.* coli* growth when related to the control. On the contrary, we observed an approx. 23% CPU decrease (100 × 25/32.5, derived from left part of Figure 6) when the final material (of a lower concentration than 1 mg/mL) vs. CTRL was allowed to interact with bacterial suspension for 2 h (subsequently, the bacterial suspension was collected from their surfaces and incubated on plate count agar culture dish overnight; for graphical scheme of our antibacterial tests; see Appendix A).

Additional antibacterial activity of the modified PET substrate with grafted trimetallic nanocomposites on top of it against *E. coli* (right part of Figure 6) can be achieved by a short (5 min) UVA light irradiation (365 nm). The slight antibacterial activity increase in the final material upon UVA light (orange column in the right part of Figure 6) is probably caused by the presence of SPION at the surface of the final material (as evidenced by XPS). SPION absorbs UVA light efficiently and is known for hydroxyl radicals’ formation [34]. Obviously, the short UVA-light irradiation also decreases the standard deviation (expressed as error bars in Figure 6), which thus stabilizes the observed mild antibacterial effect (compare right and left parts of Figure 6). We stress that a short time of UVA-light exposure needs to be used because of the protein presence within trimetallic nanocomposites. Indeed, in our previous studies [34,37], dealing with the impact of UVA light on monometallic and bimetallic nanocomposites of a similar kind, we have demonstrated the possibilities and limitations of photo-induced charge transfer processes and hydroxyl radical formation in aqueous solutions. It should also be noted that prolonged UV light is known to cause the oxidation of lipids and proteins, which is, consequently, harmful to bacteria [45].

Interestingly, the growth of CPU number is observed for pristine PET substrate when UV lamp irradiated for 5 min in comparison to CTRL (exceeding the experimental error)—compare black and red columns in right part of Figure 6. It may be caused by the change in surface wettability of the pristine PET under UVA light. Based on the literature [46], UV excimer light (172 nm) enhances the hydrophilic nature of PET film. Although we are using 365 nm, there might be still some impact on PET hydrophilicity. Consequently, slightly improved aqueous wettability of UV-irradiated PET could be considered, leading to an increased *E. coli* growth in comparison to the situation on non-UV-irradiated PET. On the other hand, plasma treatment and surface functionalization of PET by BPD resulted in decreased aqueous wettability (Figure 5) and that is probably the reason for *E. coli* growth decrease (average CPU decrease) in UV-irradiated cases of pl-PET and pl-BPD-PET in comparison to the UV-irradiated PET (right part of Figure 6). It should be, however, noted that the wettability of substrates represents only one of the key parameters influencing antibacterial results. In our case, aqueous wettability is the best for the final material (Figure 5); thus, the best bacteria growth could be expected while considering solely this parameter. However, it contradicts the results. Therefore, the composition of surfaces prevails and plays a crucial role when antibacterial effect is evaluated. The combination of iron oxide particles and/or Ag nanoparticles in the final material further increases the antibacterial effect of UV light as reported here, and correlates well with previous findings [47,48]. Thus, our results and methodology are consistent with the published scientific literature.

We hypothesize that stabilized antibacterial effects of the final material due to UVA light irradiation can be envisioned as a new approach to fight against bacteria, avoiding simultaneously potential bacterial resistance development. Further work can be devoted to optimization of PET surface coverage by the trimetallic nanocomposite and adjustment of mutual metal concentrations within the trimetallic nanocomposites to reach a higher antibacterial effect. Attempts to graft the trimetallic nanocomposites to other polymeric substrates that are intended to be surface-modified with the aim of improved antibacterial and antifouling properties can also be envisaged.

## 3. Experimental Part

### 3.1. Materials for Trimetallic Nanocomposites Synthesis

BSA (>98%), gold(III) chloride trihydrate (HAuCl_4_·3H_2_O; ≥99.9%), silver nitrate (AgNO_3_; 99.9999%), iron(III) chloride hexahydrate (FeCl_3_·6H_2_O; ≥99%), and sodium hydroxide (NaOH; ≥98.0%) were purchased from Sigma-Aldrich (St Louis, MO, USA) and used as received (without any further purification). Hydrochloric acid (35%) was purchased from Penta s.r.o. (Prague, Czech Republic). Deionized (DI) water was prepared by purging Milli-Q purified water (Millipore Corp., Bedford, MA, USA) and was used for nanocomposite synthesis.

### 3.2. AuAg-BSA-SPION@pl-BPD-PET Preparation

Polyethyleneterephthalate thin foil (PET, ρ = 1.3 g·cm^−3^, thickness 23 µm, Goodfellow Cambridge Ltd., Huntingdon, UK) was selected for the study. Firstly, samples were activated in DC Ar^+^ plasma on Balzers SCD 050 device (BalTec AG, Pfäffikon, Switzerland): exposure duration 120 s, discharge power 8.3 W, and pressure of 10 Pa. Secondly, the samples were immersed into a methanol-based solution of biphenyl-4,4′-dithiol (BPD; c = 1 × 10^−3^ mol·L^−1^; Sigma-Aldrich Corp., St Louis, MO, USA) for 24 h. Afterwards, they were transferred into the aqueous solution of AuAg-BSA-SPION nanocomposites (prepared exactly as described in the details in [33]), where they were allowed to stay for another 24 h. After taking the samples out from the nanocomposite solution, they were washed out and dried in N_2_ stream. Samples were prepared and stored under laboratory conditions. The preparation of the final material (i.e., modified PET substrates covered with nanocomposites) was repeated twice (always with a new set of nanocomposites) and all samples were prepared in hexaplicates.

### 3.3. Analytical Methods

The X-ray photoelectron spectra (XPS) were recorded on a Nexsa G2 XPS system (Thermo Fisher Scientific, Brno, Czech Republic) equipped with a monochromatic Al-Kα source and photon energy of 1486.7 eV. The spectra were taken at two angles: 0° (i.e., normal to the surface; providing general sample composition below the surface) and 81° (angle considered from the normal; more information provided directly from top surface layers), under vacuum (1.6 × 10^−9^ mbar) and at a controlled ambient temperature of 20 °C. Charge neutralization techniques were systematically switched on during the measurement. Spectral analysis and data interpretation were conducted using the Avantage software version 6.5.1 provided by Thermo Fisher Scientific.

Ultraviolet–visible (UV-Vis) spectroscopy was used to confirm the presence of AuAg nanostructures on plasma-treated PET. Absorption spectra were evaluated using a Lambda 25 spectrophotometer (PerkinElmer Inc., Shelton, CT, USA) in the spectral range of 400 to 600 nm, and with scan speed 240 nm·min^−1^ and step of 1 nm. The reference spectrum of pristine PET was subtracted from the collected spectrum of nanostructured PET. The optical absorption error values were lower than 1.5%.

Infrared spectroscopy analysis was performed by a Fourier-transform infrared (FTIR) spectrometer Nicolet iS5 (Fisher Scientific) with a diamond crystal iD7 ATR accessory. The spectra were obtained as an average from 128 measurement cycles in the spectral range of 2000–600 cm^−1^ with the data interval of 0.964 cm^−1^.

Static water contact angle (WCA) of distilled water, characterizing chemical and morphological differentiation for each modification step, was determined at laboratory conditions on four samples and at ten spots by DSA 100 goniometer (KRÜSS GmbH, Hamburg, Germany). Drops of (2.0 ± 0.2) µL were dripped on the selected samples. WCA values were determined by the ADVANCE software (version 1.17).

Scanning electron microscopy (SEM, Lyra dual beam microscope, Tescan a. s., Brno, Czech Republic) was applied for evaluation of the surface morphology of pristine PET, plasma-treated PET, PET grafted with BPD, and subsequently with AuAg-BSA-SPION nanocomposites. SEM analysis was performed in the secondary electron (SE) imaging mode with accelerating voltage of 5 kV. The magnification of the presented SEM images is 100,000× with 2 × 2 μm^2^ scan size. To prevent surface charging, samples were attached to the holder with carbon conductive tape and coated with 10 nm thick carbon layer. The carbon layer was prepared by flash evaporation method using a sputtering device, Balzers SCD 050 device (BalTec AG, Pfäffikon, Switzerland).

An atomic force microscope, Dimension ICON (AFM, Bruker Corp., Billerica, MA, USA), was employed to investigate the surface morphology and roughness of the prepared samples at nanometer level. AFM analysis was performed with ScanAsyst mode in the air using a silicon tip on a nitride cantilever with a spring constant of 0.4 N·m^−1^. The acquired data were processed by the NanoScope Analysis software (version 1.40) to obtain the average surface roughness values (*R_a_*). The size of the scans was 300 × 300 nm^2^.

### 3.4. Antibacterial Effect

The antibacterial effect of the final material (AuAg-BSA-SPION nanocomposites grafted on plasma-treated BPD-modified PET) was tested using the drop plate method. The measurement was carried out using the Gram-negative bacteria (G-) *Escherichia coli* (*E. coli*; DBM 3138). One colony was transferred to 25 mL of Luria–Bertani (LB) liquid medium and incubated overnight on the orbital shaker at 37 °C to prepare bacterial inoculum. Next day, the inoculum was diluted in phosphate-buffered saline (PBS) to final concentration, which was approx. 1.3 × 10^3^ colony-producing units (CPUs) per mL (*E. coli*/mL). This bacterial suspension was dropped (150 μL) on top of a sample surface placed in PS Petri dish (each sample in triplicates). As a control (CTRL), a Petri dish bottom (without any sample inserted) was used. Samples intended for irradiation were placed under a UV lamp (365 nm), 8 cm away from the surface, and for five min. After 2 h, five drops (25 μL) of bacterial suspension from each sample surface were placed on a plate count agar culture dish (PCA, Oxoid, containing tryptone 5 g·L^−1^, yeast extract 2.5 g·L^−1^, glucose 1 g·L^−1^, agar 15 g·L^−1^). The cultivation was carried out overnight at room temperature. The next day, the grown CPUs were counted. The measured counts can be compared with the maximal theoretical value of CPUs that can be calculated for the CTRL. For example, 0.025 × 1300 CPU, i.e., 32,5 CPU. All experiments were performed under sterile conditions.

The results are depicted as the average of CPUs from all 15 drops from one type of sample surface (five drops from each triplicate) with 95% confidence interval. To show the differences between the samples and the CTRL, the two-sample *t*-test assuming equal variance (Excel, Microsoft Office, Redmond, WA, USA) was performed.

## 4. Conclusions

The trimetallic nanocomposites consisting of AuAg nanoclusters and superparamagnetic iron oxide nanoparticles were successfully grafted on modified PET substrates as evidenced by spectroscopic and microscopic characterization. This newly developed material (AuAg-BSA-SPION@pl-BPD-PET) was proved by the colony-producing assay to be efficient against *E. coli* growth. Short-time (5 min) UVA light (365 nm) exposure induced further antibacterial effect of the newly developed material. It is the first trimetallic nanocomposite grafted on a modified PET substrate that is successfully used in *E. coli* growth regulation. It combines the effects of small noble metal nanostructures containing Ag and UVA light-activated iron oxide particles.

## Data Availability

The data supporting this article has been included as part of Appendix A. Further inquiries can be directed to the corresponding authors.

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
