# Peer review of "Trimetallic Nanocomposites Grafted on Modified PET Substrate Revealing Antibacterial Effect Against Escherichia coli"

_molecules, 2025, doi:10.3390/molecules30244820_

Round 1

Reviewer 1 Report

Comments and Suggestions for Authors

The authors prepared trimetallic nanocomposites modified PET substrate and discussed their antibacterial effect under UV treated. However, some issues should be addressed before publication.

  1. About the cost of AuAg-BSA-SPION nanocomposites, the high cost of Au and Ag would increase the price of modified PET, how to control the performance/cost ratio of modified PET?
  2. Table 1 should be placed after the XPS results. Or else, where are these results from?
  3. In Figure 1, the Figure caption (A to F) should be placed on the top-left corner, it is hard to find them now. The general XPS spectrum should be provided in Figure 1.
  4. The manuscript has only 4 Figures, so the SEM images in Figure SI-4 should be placed before the AFM pictures (Figure 2).
  5. From Figure 4, what is difference between the CTRL and PET. After illuminated for 5 min, why the CPU of PET is higher than that with no treated. While the CPU of CTRL and AuAg-PET decrease.

Author Response

Comments and Suggestions for Authors

The authors prepared trimetallic nanocomposites modified PET substrate and discussed their antibacterial effect under UV treated. However, some issues should be addressed before publication.

  1. About the cost of AuAg-BSA-SPION nanocomposites, the high cost of Au and Ag would increase the price of modified PET, how to control the performance/cost ratio of modified PET?

Answer: We thank to the reviewer for this question. Despite the high cost of noble metals, it should be reminded that there is only 0.75 mg/mL of Au and 0.12 mg/mL of Ag in the concentrated variants of nanocomposite solutions as determined by ICP-MS in our previous work (Svačinová et al., J. Mater. Chem. B 2024). Considering the 0.1 atomic % of Au grafted on pl-BPD-PET surface as evidenced by XPS in the present work (Table 1), it can be concluded that the price for the new material fabrication is not enormous since there is only very tiny amount of Au and Ag.

2. Table 1 should be placed after the XPS results. Or else, where are these results from?

Answer: We have moved XPS survey scans shown in Figures SI-3A, SI-3B, and SI-3C from the Supporting Information into the main text. Therefore, it is newly denoted as Figures 1A, 1B, and 1C. Therefore, the other figures had to be renumbered.

We have also included a few sentences in the main text so that it is understandable where the data in Table 1 comes from. The text placed immediately after new Figure 1 reads as follows: Based on survey scans and XPS signal analyses, atomic percentage of selected elements can be determined and is listed in Table 1 for the trimetallic nanocomposite dropped and/or grafted on substrates.”  

3. In Figure 1, the Figure caption (A to F) should be placed on the top-left corner, it is hard to find them now. The general XPS spectrum should be provided in Figure 1.

Answer: We have moved XPS survey scans shown in Figures SI-3A, SI-3B, and SI-3C from the Supporting Information into the main text. Therefore, it is newly denoted as Figures 1A, 1B, and 1C. Therefore, the other figures had to be renumbered.

Therefore, caption A-H in current Figure 2 (previously Figure 1) has been moved to the top-left corner as required by the reviewer. Moreover, Figures 2G and 2H containing the detailed XPS scans of S 2p region have been added (they were unintentionally missing in the previous version of the manuscript).

4. The manuscript has only 4 Figures, so the SEM images in Figure SI-4 should be placed before the AFM pictures (Figure 2).

Answer: We have moved SEM images from the SI file into the main text and numbered it as Figure 3. Simultaneously, the other images have been renumbered.

5. From Figure 4, what is difference between the CTRL and PET. After illuminated for 5 min, why the CPU of PET is higher than that with no treated. While the CPU of CTRL and AuAg-PET decrease.

Answer: We have reformulated the text about antibacterial testing in Experimental section as well as improved figure caption of the appropriate figure (previously Figure 4, currently Figure 6). We have included a new paragraph to comment and explain the question posed by the reviewer. The paragraph reads as follows and can be found out in section 3.4: “Interestingly, the growth of CPU number is observed for pristine PET substrate when UV lamp irradiated for 5 minutes in comparison to CTRL (exceeding the experimental error) – compare black and red columns in right part of Figure 6. It may be caused by the change of surface wettability of the pristine PET under UVA light. Based on the literature 47, UV excimer light (172 nm) enhances the hydrophilic nature of PET film. Although we are using 365 nm, there might be still some impact on PET hydrophilicity. Consequently, slightly improved aqueous wettability of UV-irradiated PET could be considered, leading to an increased E. coli growth in comparison to the situation on non-UV-irradiated PET. On the other hand, plasma treatment and surface functionalization of PET by BPD resulted in a decreased aqueous wettability (Figure 5) that is probably the reason for E. coli growth decrease (average CPU decrease) in UV-irradiated cases of pl-PET and pl-BPD-PET in comparison to the UV-irradiated PET (right part of Figure 6). It should be, however, reminded that the wettability of substrates represents only one of key parameters influencing antibacterial results. In our case, aqueous wettability is the best for the final material (Figure 5), thus, the best bacteria growth could be expected while considering solely this parameter. However, it contradicts the results. Therefore, the composition of surfaces prevails and plays a crucial role when antibacterial effect is evaluated. Combination of iron oxide particles and/or Ag nanoparticles in the final material further increases the antibacterial effect of UV light as reported here, and correlates well with previous findings 48 49. Thus, our results and methodology are consistent with the published scientific literature.”

We thank to the reviewer for his/her valuable comments and inspiring questions that helped to improve the quality of our manuscript.

Reviewer 2 Report

Comments and Suggestions for Authors

The article by Veronika Lacmanova et al "Trimetallic nanocomposites grafted on modified PET substrate revealing antibacterial effect against Escherichia Coli" is devoted to the urgent problem of creating antibacterial coatings on polymer substrates using trimetallic nanocomposites (AuAg-BSA-SPION). The authors demonstrate a multi-step approach to PET surface modification, including plasma treatment, chemical binding via dithiol linker (BPD), and immobilization of nanocomposites. Antibacterial effect against E. coli has been confirmed experimentally, including under UV irradiation conditions. The work is well structured, but requires additional clarifications and clarifications in a number of sections. The work is of considerable interest, but contains a number of points that require further elaboration and clarification.

1) The introduction provides a review of the literature on metal-containing nanomaterials, but there is a lack of clear justification for the novelty of the AuAg-BSA-SPION trimetallic system in comparison with known bi- and trimetallic analogues. It is advisable to formulate the research hypothesis and the expected synergetic effect more clearly. It is recommended to transfer Scheme 1 and description of the synthesis of AuAg-BSA-SPION@pl-BPD-PET material to the "results and discussion" section.

2) it is better to replace the term device with the term material

3) In the description of the antibacterial effect, it is necessary to add a comparison with the previously described effects of materials based on three-metal composites.

4) It is not clear why the growth of colony producing units for the PET sample occurs during UV irradiation, because UV is known to have an antimicrobial effect.

5) It is necessary to add a discussion about the possible mechanism of action, whether it is ion-mediated or ROS-related.

6) There are typos and stylistic inaccuracies in the text (for example, "pl-BPD-PET" is sometimes written without hyphens).

The article is a completed study demonstrating the creation of an antibacterial coating based on trimetallic nanocomposites on a flexible PET substrate. The work has practical value and may be of interest to specialists in the field of nanomaterials, antimicrobial coatings and polymer modification. The article can be accepted for publication after the removal of these comments

Author Response

Comments and Suggestions for Authors

The article by Veronika Lacmanova et al "Trimetallic nanocomposites grafted on modified PET substrate revealing antibacterial effect against Escherichia Coli" is devoted to the urgent problem of creating antibacterial coatings on polymer substrates using trimetallic nanocomposites (AuAg-BSA-SPION). The authors demonstrate a multi-step approach to PET surface modification, including plasma treatment, chemical binding via dithiol linker (BPD), and immobilization of nanocomposites. Antibacterial effect against E. coli has been confirmed experimentally, including under UV irradiation conditions. The work is well structured, but requires additional clarifications and clarifications in a number of sections. The work is of considerable interest, but contains a number of points that require further elaboration and clarification.

1) The introduction provides a review of the literature on metal-containing nanomaterials, but there is a lack of clear justification for the novelty of the AuAg-BSA-SPION trimetallic system in comparison with known bi- and trimetallic analogues. It is advisable to formulate the research hypothesis and the expected synergetic effect more clearly. It is recommended to transfer Scheme 1 and description of the synthesis of AuAg-BSA-SPION@pl-BPD-PET material to the "results and discussion" section.

Answer: In the revised version of our manuscript, we have tried to reformulate the research hypothesis and to be clearer about the novelty in introduction. The text of introduction concerning the novelty statement (underlined in the cited text) now reads as it follows: “Here, we would like to employ AuAg nanostructures combined with iron oxide particles within one nanocomposite (further on called as the trimetallic nanocomposite and abbreviated as AuAg-BSA-SPION), chemically attach these trimetallic nanocomposite on modified PET substrates and then test the antibacterial properties of the final material (AuAg-BSA-SPION@pl-BPD-PET). The first novelty of the present work lies in the preparation of the final material per se. In the next step, we want to evaluate the effect of the final material against Escherichia Coli for two reasons: (i) it has been proven that hydroxyl radicals are effective in E. Coli growth inhibition27, and (ii) the results can be compared with the previous work dealing with grafted bimetallic nanocomposites serving as antibacterial PET coatings against E. Coli 26. It is expected that due to the presence of Ag within the trimetallic nanocomposites, the final material will provide a more pronounced antibacterial effect than that obtained for the case of the grafted bimetallic nanocomposites (Au-BSA-SPION, our recent work 26) because it is known that inhibitory action against gram-negative bacteria is higher for nanosilver than for any other metallic nanoparticles 28.

According to the literature 26 29, the optimized plasma treatment of PET (representing a flexible, light, UVA-transparent polymeric substrate) has been selected to change its inert character. The optimal procedure of plasma treatment can not only activate the surface of PET, while its negligible degradation occurs, but, simultaneously, any dirtiness can be removed from PET surface. Subsequently, a dithiol linker (BPD) could be employed to graft the trimetallic nanocomposites on the modified PET surface. From one of our previously published works 30, we know that BPD represents a “rigid” linker and as per se, it is conveniently oriented on the PET surface with one of its two sulfhydryl groups pointing outside the surface. This can enable noble metal nanostructures to bond via the covalent interaction between Au/Ag with S, similarly, as evidenced for AuNPs and/or AgNPs in refs 31 32.

Importantly, the trimetallic nanocomposites grafted in this work were synthesized by the green chemistry (protein-templated) approach that has been developed by us recently33. The trimetallic nanocomposites consist of two kinds of functional nanostructures: AuAg irregularly shaped nanoclusters (diameters below 2 nm) embedded within the protein (BSA) and SPION (superparamagnetic iron oxide nanoparticles of sizes well below 8 nm) attached to the same protein 33. These two kinds of nanostructures are positioned at different binding sites of the protein since Au and Ag prefer the interaction with sulfhydryl groups (mostly cysteine residues); while iron binds to N-terminal region of the protein 34 and may prefer to interact with carboxylates, oxygen and/or nitrogen-terminated functional groups. Therefore, we expect that AuAg nanostructures embedded in BSA matrix can be bonded via the “free” sulfhydryl group of BPD to PET substrate, enabling then the attachment of the whole trimetallic nanocomposite. Consequently, the steric arrangement of the trimetallic nanocomposite can change while grafting to the surface, leading to new properties that have not been observed so far, e.g. antibacterial properties.

Moreover, it is known from the literature 35 that the binding mode is one of key determinants for the antimicrobial performance of iron oxide/silver nanocomposites. It should be reminded that the trimetallic nanocomposites, developed by us33, revealed biocompatibility with healthy cell lines (RPE-1) despite the presence of Ag (till 0.12 mg/mL). Nevertheless, the biocompatibility of these trimetallic nanocomposites towards cells may change due to potential conformational changes of the protein matrix, induced because of their grafting on PET substrate. Furthermore, it is important to investigate the impact of UVA light exposure for short times (5 min) on antibacterial properties of the final material because of the presence of SPION (as inspired by the recent and previous works 26 36). However, any prolonged UVA-light irradiation should be avoided because it is well-known to cause irreversible changes to the nanocomposites containing noble metal nanostructures embedded in BSA 37. Therefore, a direct comparison of antibacterial properties of the final material under no vs UVA light irradiation represents another novelty of this work. “

 We moved Scheme 1 into the beginning of Results and discussion section. Similarly, the sentence referring to it.

2) it is better to replace the term device with the term material

Answer: We have replaced the term as requested throughout the whole manuscript.

3) In the description of the antibacterial effect, it is necessary to add a comparison with the previously described effects of materials based on three-metal composites.

Answer: We tried to improve our discussion of results of antibacterial effect and added a new paragraph in section 3.4. This paragraph now reads as follows: “It is quite hard to directly compare the results of antibacterial testing of the final material under study with any other three-metal composite published so far in the scientific literature because different methods of antibacterial effect evaluation were employed, such as agar disc diffusion test 21, well diffusion assay  23, macro broth dilution approach25 for instance. The only available article using drop plate method and reporting on antibacterial properties against E. coli has dealt with CuO-NiO-ZnO nanoparticles prepared by a co-precipitation method 24. Considering (i) tested concentration range (1-10 mg/mL) of CuO-NiO-ZnO nanoparticles and (ii) investigated time of bacterial growth (2-11 h), these three-metal oxidic nanoparticles 24 revealed a negligible antibacterial effect in comparison to the final material under study in the present work. Indeed, their 24 lowest concentrations of CuO-NiO-ZnO (1 mg/mL) when allowed to interact with E. coli for 11 hours, manifested itself by an approx. 14 % decrease of E. coli growth when related to the control. On the contrary, we observed approx. 23 % CPU decrease (100×25/32.5, derived from left part of Figure 6) when the final material vs CTRL allowed to interact with bacterial suspension for 2h (subsequently, the bacterial suspension collected from their surfaces and incubated on plate count agar culture dish overnight – for graphical scheme of our antibacterial tests see Figure SI-4 in Supporting Information).”

4) It is not clear why the growth of colony producing units for the PET sample occurs during UV irradiation, because UV is known to have an antimicrobial effect.

Answer: We thank to the reviewer for this remark. Indeed, it is necessary to distinguish between the types of UV irradiation. While UVC kills bacteria easily, UVA is not that efficient. Our results clearly showed that UVA light had no significant impact on E. coli in the control (CTRL) – please, compare black columns on right and left parts of Figure 6 in the revised version of the manuscript (previously the figure denoted as Figure 4).  In fact, the results are comparable within the experimental error. However, it is true that there is a non-negligible increase in CPU number (exceeding the experimental error) when pristine PET irradiated by UVA and antibacterial test performed. It may be caused by the change of surface wettability of the pristine PET under UVA light. Therefore, we have included a new paragraph regarding this issue in section 3.4: “Interestingly, the growth of CPU number is observed for pristine PET substrate when UV lamp irradiated for 5 minutes in comparison to CTRL (exceeding the experimental error) – compare black and red columns in right part of Figure 6. It may be caused by the change of surface wettability of the pristine PET under UVA light. Based on the literature 44, UV excimer light (172 nm) enhances the hydrophilic nature of PET film. Although we are using 365 nm, there might be still some impact on PET hydrophilicity. Consequently, slightly improved aqueous wettability of UV-irradiated PET could be considered, leading to an increased E. coli growth in comparison to the situation on non-UV-irradiated PET. On the other hand, plasma treatment and surface functionalization of PET by BPD resulted in a decreased aqueous wettability (Figure 5) that is probably the reason for E. coli growth decrease (average CPU decrease) in UV-irradiated cases of pl-PET and pl-BPD-PET in comparison to the UV-irradiated PET (right part of Figure 6). It should be, however, reminded that the wettability of substrates represents only one of key parameters influencing antibacterial results. In our case, aqueous wettability is the best for the final material (Figure 5), thus, the best bacteria growth could be expected while considering solely this parameter. However, it contradicts the results. Therefore, the composition of surfaces prevails and plays a crucial role when antibacterial effect is evaluated. “

5) It is necessary to add a discussion about the possible mechanism of action, whether it is ion-mediated or ROS-related.

Answer: Prompted by this reviewer comment, we have added a new paragraph in section 3.4 and included new references as well. The paragraph reads as follows: “Since we do not expect antibacterial activity against E. coli to be induced by the protein (BSA), neither by eco-friendly prepared Au nanostructures 42 and/or SPION present within the final material 26, the antibacterial effect is most probably induced by the presence of Ag (despite its very low concentration at the surface, 0.2 % derived from XPS data, Table 1). Ag nanoparticles and released Ag ions are known to interact with biomolecules, consequently resulting in a detrimental effect on bacteria 43. Additionally, radical oxygen species formation induced by Ag nanoparticles may be considered as another mechanism of its antibacterial action 43, 44. The assumption about Au, SPION and BSA negligible impact on E. coli growth can be well documented by referring to our previous work dealing with Au-BSA-SPION composites grafted on surface modified PET substrate (Au-BSA-SPION@pl-BPD-PET) 26. Albeit there are scientific articles about bactericidal effect of Au nanoparticles on E. coli 45, the decrease of Au size to nanoclusters (below 2 nm in diameter) and their entrapment in the protein matrix, may be reasons for no antibacterial effect of Au-BSA-SPION@pl-BPD-PET on E. coli 26 42.”

6) There are typos and stylistic inaccuracies in the text (for example, "pl-BPD-PET" is sometimes written without hyphens).

Answer: We went through the whole main text and checked all abbreviations, especially those containing pl-BPD-PET. It should be now correct and consistent.

The article is a completed study demonstrating the creation of an antibacterial coating based on trimetallic nanocomposites on a flexible PET substrate. The work has practical value and may be of interest to specialists in the field of nanomaterials, antimicrobial coatings and polymer modification. The article can be accepted for publication after the removal of these comments

We thank to the reviewer for his/her valuable comments and inspiring questions that helped to improve the quality of our manuscript.